# EWoK: Tackling Robust Markov Decision Processes via Estimating Worst Kernel

## Abstract

Robust Markov Decision Processes (RMDPs) provide a framework for sequential decision-making that is robust to perturbations on the transition kernel. However, current RMDP methods are often limited to small-scale problems, hindering their use in high-dimensional domains. To bridge this gap, we present **EWoK**, a novel approach for the online RMDP setting that **E**stimates the **Wo**rst transition **K**ernel to learn robust policies. Unlike previous works that regularize the policy or value updates, EWoK achieves robustness by simulating the worst scenarios for the agent while retaining complete flexibility in the learning process. Notably, EWoK can be applied on top of any off-the-shelf *non-robust* RL algorithm, enabling easy scaling to high-dimensional domains. Our experiments, spanning from simple Cartpole to high-dimensional MinAtar and DeepMind Control Suite environments, demonstrate the effectiveness and applicability of the EWoK paradigm as a practical method for learning robust policies.

## 1 Introduction

In reinforcement learning (RL), we are concerned with learning good policies for sequential decision-making problems modeled as Markov Decision Processes (MDPs) (Puterman, 1994; Sutton & Barto, 2018). MDPs assume that the transition model of the environment is fixed across training and testing, but this is often violated in practical applications. For example, when deploying a simulator-trained robot in reality, a notable challenge is the substantial disparity between the simulated environment and the intricate complexities of the real world, leading to potential subpar performance upon deployment. Such a mismatch may significantly degrade the performance of the trained policy in testing. To deal with this issue, the robust MDP (RMDP) framework has been introduced in (Iyengar, 2005; Nilim & El Ghaoui, 2005; Wiesemann et al., 2013), aiming to learn policies that are robust to perturbation of the transition model within an uncertainty set.

Existing works on learning robust policies in RMDPs often suffer from poor scalability to high-dimensional domains. Specifically, model-based methods that solve RMDPs (Wiesemann et al., 2013; Ho et al., 2020; Behzadian et al., 2021; Derman et al., 2021; Grand-Clément & Kroer, 2021; Kumar et al., 2022b) require access to the nominal transition probability, making it difficult to scale beyond tabular settings. While some recent works (Wang et al., 2022; Wang & Zou, 2022b; Kumar et al., 2022b; 2023) introduce model-free methods that add regularization to the learning process, the effectiveness of their methods is not validated in high-dimensional environments. In addition, these methods are based on particular RL algorithms (*e.g.*, policy gradient, Q learning), limiting their general applicability. We defer a more detailed discussion on related works to Section 5.

In this work, we tackle the problem of learning robust policies in RMDPs from an alternative direction. As shown in Figure 1, unlike previous works that explicitly regularize the learning process, we propose to approximately sample next states from an **e**stimated **wo**rst transition **k**ernel (EWoK) while leaving the RL part untouched. In RMDPs, a worst transition kernel is one within the uncertainty set that leads to the minimal possible return (see Definition 3.1). Intuitively, EWoK aims to situate the agent in the worst scenarios for learning policies robust to perturbations. It can be applied on top of any (deep) RL algorithm, offering good scalability to high-dimensional domains.

Specifically, EWoK is built upon our theoretical insights into the relationship between a worst transition kernel and the nominal one. For the KL-regularized uncertainty set considered in our work, a

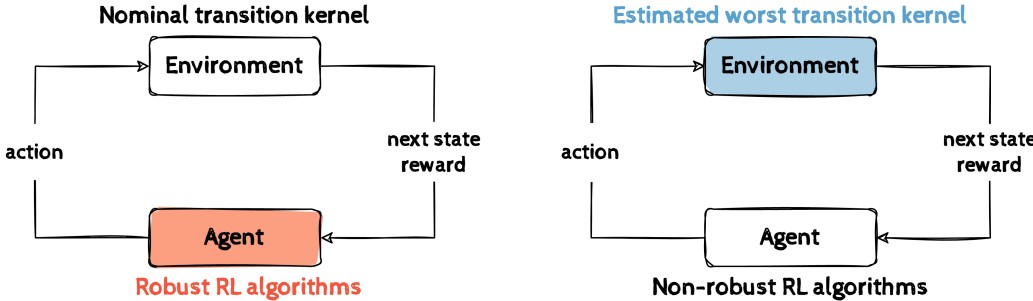

Figure 1: The agent-environment interaction loop during training. **Left**: Existing methods typically regularize how an agent updates its policy to improve robustness. **Right**: Our work approximates a worst transition kernel, so the agent essentially learns its policy under the worst scenarios and can use any non-robust RL algorithm.

worst kernel essentially modifies the next-state transition probability of the nominal kernel, discouraging the transitions to states with higher values while encouraging transitions to lower-value states. Using this connection, we are able to sample the next states such that they are approximately distributed according to the worst transition probability. We establish the convergence of the estimated worst kernel to the true worst kernel and present a practical algorithm suitable for high-dimensional domains.

To verify the effectiveness of our method, we conduct experiments on multiple environments ranging from small-scale classic control tasks to high-dimensional MinAtar games (Young & Tian, 2019) and DeepMind Control tasks (Tunyasuvunakool et al., 2020). The agent is trained in the nominal environment and tested in environments with perturbed transitions. Since our method is agnostic to the underlying RL algorithm, we can easily plug it into a Double-DQN (van Hasselt et al., 2016) agent for discrete-action environments or a SAC (Haarnoja et al., 2018a) agent for continuous-action environments. Experiment results demonstrate that with our method, the learned policy suffers from less performance degradation when the transition kernel is perturbed.

In summary, our paper makes the following contributions:

- To learn robust policies in RMDPs, we propose to approximately simulate the "worst" transition kernel, rather than regularizing the learning process. This opens up a new paradigm for learning robust policies in RMDPs.

- We theoretically characterize the "worst" kernel under the KL uncertainty set, which is amenable to approximate simulation for environments with large state spaces.

- Our method is not tied to a particular RL algorithm and can be easily integrated with any deep RL method. This flexibility translates to the good scalability of our method in complex high-dimensional domains such as MinAtar and DeepMind Control. To the best of our knowledge, our work is the first that enjoys such flexibility among related works in RMDPs.

## 2 PRELIMINARIES

*Notations.* For a finite set $\mathcal{Z}$, we write the probability simplex over it as $\Delta_{\mathcal{Z}}$. Given two real functions $f, g : \mathcal{Z} \to \mathbb{R}$, their inner product is $\langle f, g \rangle = \sum_{z \in \mathcal{Z}} f(z)g(z)$. For distributions $P, Q$, we denote the Kullback–Leibler (KL) divergence of $P$ from $Q$ by $D_{\mathrm{KL}}(P \,\|\, Q)$.

### 2.1 MARKOV DECISION PROCESSES

A Markov decision process (MDP) (Sutton & Barto, 2018; Puterman, 1994) is a tuple $(\mathcal{S}, \mathcal{A}, P, R, \gamma, \mu)$, where $\mathcal{S}$ and $\mathcal{A}$ are the state space and the action space respectively, $P : \mathcal{S} \times \mathcal{A} \to \Delta_{\mathcal{S}}$ is the transition kernel, $R : \mathcal{S} \times \mathcal{A} \to \mathbb{R}$ is the reward function, $\gamma \in [0, 1)$ is the discount factor, and $\mu \in \Delta_{\mathcal{S}}$ is the initial state distribution. A stationary policy $\pi : \mathcal{S} \to \Delta_{\mathcal{A}}$ maps a state to a probability distribution over $\mathcal{A}$. We use $P(\cdot|s, a) \in \Delta_{\mathcal{S}}$ to denote the probabilities of transiting to

the next state when the agent takes action $a$ at state $s$. For a policy $\pi$, we denote the expected reward and transition by:

$$R^\pi(s) = \sum_a \pi(a|s)R(s,a), \quad P^\pi(s'|s) = \sum_a \pi(a|s)P(s'|s,a). \tag{1}$$

The value function $v^\pi : \mathcal{S} \to \mathbb{R}$ maps a state to the expected cumulative reward when the agent starts from that state and follows policy $\pi$, *i.e.*,

$$v^\pi(s) = \mathbb{E}\left[\sum_{t=0}^\infty \gamma^t R(s_t, a_t) \,\middle|\, s_0 = s, a_t \sim \pi(\cdot|s_t), s_{t+1} \sim P(\cdot|s_t, a_t)\right]. \tag{2}$$

It is known that $v^\pi$ is the unique fixed point of the Bellman operator $T^\pi : T^\pi v = R^\pi + \gamma P^\pi v$ (Puterman, 2014). The agent's objective is to obtain a policy $\pi^*$ that maximizes the discounted return

$$J^\pi = \mathbb{E}\left[\sum_{t=0}^\infty \gamma^t R(s_t, a_t) \,\middle|\, s_0 \sim \mu, a_t \sim \pi(\cdot|s_t), s_{t+1} \sim P(\cdot|s_t, a_t)\right] = \langle \mu, v^\pi \rangle. \tag{3}$$

## 2.2 ROBUST MARKOV DECISION PROCESSES

In MDPs, the system dynamic $P$ is usually assumed to be constant over time. However, in real-life scenarios, it is subject to perturbations, which may significantly impact the performance in deployment (Mannor et al., 2007). Robust MDPs (RMDPs) provide a theoretical framework for taking such uncertainty into consideration, by taking $P$ as not fixed but chosen adversarially from an uncertainty set $\mathcal{P}$ (Iyengar, 2005; Nilim & El Ghaoui, 2005). Since we may consider different dynamics $P$ in the RMDPs context, in the following, we will use subscript $P$ to make the dependency explicit. The objective in RMDPs is to obtain a policy $\pi^*_{\mathcal{P}}$ that maximizes the robust return

$$J^\pi_{\mathcal{P}} = \min_{P \in \mathcal{P}} J^\pi_P. \tag{4}$$

However, solving RMDPs for general uncertainty sets is NP-hard while an optimal policy can be non-stationary (Wiesemann et al., 2013). To make RMDPs tractable, we need to make some assumptions about the uncertainty set.

## 2.3 RECTANGULAR UNCERTAINTY SET

One commonly used assumption to enable tractability for RMDPs is rectangularity. Specifically, we assume that the uncertainty set $\mathcal{P}$ can be factorized over states-actions:

$$\mathcal{P} = \underset{(s,a)\in(\mathcal{S}\times\mathcal{A})}{\times} \mathcal{P}_{sa}, \tag{sa-rectangularity}$$

where $\mathcal{P}_{sa} \subseteq \Delta_{\mathcal{S}}$. In other words, the uncertainty in one state-action pair is independent of that in another state-action pair.

Under this assumption, RMDPs will admit a deterministic optimal policy as in the standard MDPs (Iyengar, 2005; Nilim & El Ghaoui, 2005). The rectangularity assumption also allows the robust value function to be well-defined:

$$v^\pi_{\mathcal{P}} = \min_{P \in \mathcal{P}} v^\pi_P, \quad \text{and} \quad v^*_{\mathcal{P}} = \max_\pi v^\pi_{\mathcal{P}}. \tag{5}$$

In addition, $v^\pi_{\mathcal{P}}$ and $v^*_{\mathcal{P}}$ are the unique fixed points of the robust Bellman operator $T^\pi_{\mathcal{P}}$ and the optimal robust Bellman operator $T^*_{\mathcal{P}}$ respectively, which are defined as

$$T^\pi_{\mathcal{P}} v(s) = \min_{P \in \mathcal{P}} T^\pi_P v(s) \quad \text{and} \quad T^*_{\mathcal{P}} v(s) = \max_\pi T^\pi_{\mathcal{P}} v(s). \tag{6}$$

To model perturbations on the environment dynamics, the rectangular uncertainty set is often constructed to be centered around a nominal kernel $\bar{P}$. Since we want to measure the divergence between probability distributions, it is natural to use KL divergence (Panaganti & Kalathil, 2022; Xu et al., 2023; Shi & Chi, 2022), *i.e.*,

$$\mathcal{P}_{sa} = \{P_{sa} \mid D_{\mathrm{KL}}(P_{sa} \,\|\, \bar{P}_{sa}) \leq \beta_{sa}\}. \tag{KL uncertainty set}$$

Here $P_{sa}$ is a shorthand for $P(\cdot|s,a)$ and $\beta_{sa}$ is the uncertainty radius that controls the level of perturbation.

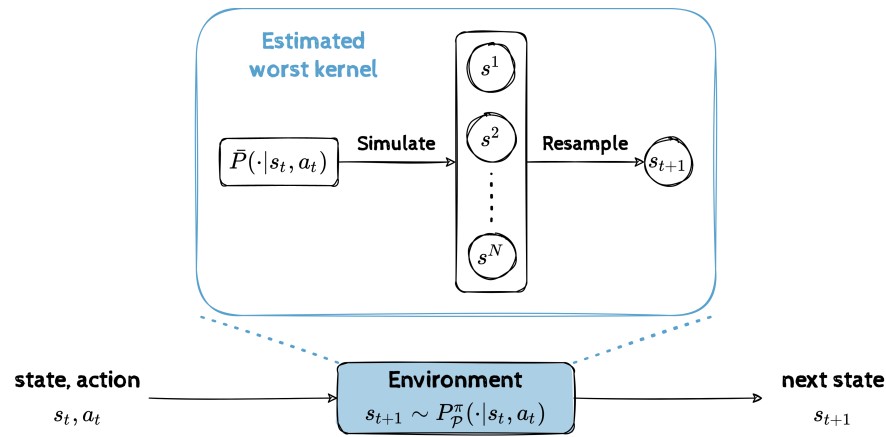

Figure 2: An illustration of how next states are sampled in the approximated worst kernel.

## 3 METHOD

As introduced earlier, our work proposes to learn robust policies by approximately simulating the *worst transition kernel*, which is defined as the one within the uncertainty set that achieves minimal robust return:

**Definition 3.1.** For an uncertainty set $\mathcal{P}$, a worst kernel for a policy $\pi$ is defined as

$$P_{\mathcal{P}}^{\pi} \in \arg\min_{P \in \mathcal{P}} J_P^{\pi}. \tag{7}$$

Training policies under this worst kernel will give us a robust policy with respect to the uncertainty set. Note that $P_{\mathcal{P}}^{\pi}$ itself is nothing more than a regular transition kernel. Learning a policy under $P_{\mathcal{P}}^{\pi}$ is no different from the standard MDP setting and we can adopt any non-robust RL algorithms to solve it. The challenge is how to approximately simulate this worst kernel $P_{\mathcal{P}}^{\pi}$. For a general uncertainty set $\mathcal{P}$, it requires an additional minimization process to find a worst kernel and it is also unclear how we can parameterize and learn $P_{\mathcal{P}}^{\pi}$ effectively.

To tackle this challenge, we characterize the connection between the nominal transition kernel and a worst one. With such a connection, we are able to obtain the next states that are approximately distributed according to $P_{\mathcal{P}}^{\pi}(\cdot|s, a)$, by properly resampling the next states from the nominal kernel (Figure 2). Formally, the following theorem describes this connection. All proofs are deferred to the appendix.

**Theorem 3.2.** *For a KL uncertainty set $\mathcal{P}$, a worst kernel is related to the nominal kernel through:*

$$P_{\mathcal{P}}^{\pi}(s'|s, a) = \bar{P}^{\pi}(s'|s, a)e^{-\delta^{\pi}(s')}, \tag{8}$$

*where $\delta^{\pi}$ is of the form*

$$\delta^{\pi}(s') = \frac{v_{\mathcal{P}}^{\pi}(s') - \omega_{sa}}{\kappa_{sa}}, \tag{9}$$

*and satisfies*

$$\sum_{s'} \bar{P}^{\pi}(s'|s, a)e^{-\delta^{\pi}(s')} = 1, \quad \sum_{s'} \bar{P}^{\pi}(s'|s, a)e^{-\delta^{\pi}(s')}(-\delta^{\pi}(s')) = \beta_{sa}. \tag{10}$$

Here, $\omega_{sa}$ and $\kappa_{sa}$ are implicitly defined by Eqn. (10). While they do not have closed forms, we can view $\omega_{sa}$ as a threshold, which encourages transitions to states with robust values lower than $\omega_{sa}$ (*i.e.*, $\delta^{\pi}(s') < 0$) and discouraging transitions to states with higher robust values. $\kappa_{sa}$ works as a temperature parameter to control how much we discourage/encourage transitions to states with high/low robust values. More specifically, we have the following proposition for the relationship between $\omega_{sa}$ and $\kappa_{sa}$ and the uncertainty radius $\beta_{sa}$.

**Proposition 3.3.** *$\omega_{sa}$, $\kappa_{sa}$ and $\beta_{sa}$ satisfy*

$$\omega_{sa} = \langle P_{\mathcal{P}}^{\pi}(\cdot|s, a), v_{\mathcal{P}}^{\pi}\rangle + \beta_{sa}\kappa_{sa}, \tag{11}$$

---

**Algorithm 1** EWoK - Learning robust policy by Estimating Worst Kernel

---

**Input**: sample size $N$, robustness parameter $\kappa$
**Initialize**: initial state $s_0$, policy $\pi$ and value function $v$, data buffer

1: **for** $t = 0, 1, 2, \cdots$ **do**
2:      Play action $a_t \sim \pi(\cdot|s_t)$.
3:      Simulate next state $s^i \sim \bar{P}(\cdot|s_t, a_t), i = 1, \cdots, N$, with the nominal environment dynamic.
4:      Choose $s_{t+1} = s^i$ with probability proportional to $e^{-\frac{v(s^i) - \frac{1}{N}\sum_{i=1}^{N} v(s^i)}{\kappa}}$.
5:      Add $(s_t, a_t, s_{t+1})$ to the data buffer.
6:      Train $\pi$ and $v$ with data from the buffer using any non-robust RL method.
7: **end for**

**Output**: a robust policy $\pi$

---

Based on theoretical results, we arrive at a method to approximately simulate a worst kernel. As illustrated in Figure 2, we first draw a batch of states from the nominal kernel $\bar{P}(\cdot|s, a)$ and then resample the next state with probability proportional to $e^{-\delta^\pi(s')}$. In this way, the next states will be approximately distributed according to $P_\mathcal{P}^\pi(\cdot|s, a)$. In practical implementations, we approximate $\delta^\pi(s')$ by

$$\hat{\delta}^\pi(s') = \frac{v(s') - \frac{1}{N}\sum_{i=1}^{N} v(s^i)}{\kappa}, \tag{12}$$

where $v$ is the robust value function approximated with neural networks, and $\kappa$ is a hyperparameter controlling the robustness level. We implement the threshold $\omega$ as the average value, a choice supported by the following proposition.

**Proposition 3.4.** $\omega_{sa}$ *can be bounded as follows,*

$$\langle P_\mathcal{P}^\pi(\cdot|s, a), v_\mathcal{P}^\pi \rangle \leq \omega_{sa} \leq \langle \bar{P}^\pi(\cdot|s, a), v_\mathcal{P}^\pi \rangle. \tag{13}$$

As the $N$ next states are sampled from the nominal kernels, we are essentially approximating the upper bound of $\omega_{sa}$ and use it as a proxy to compute $\hat{\delta}^\pi(s')$. Putting it together, we summarize our method in Algorithm 1.

**Convergence.** The core of our method is the estimation of a worst transition kernel. In practice, however, we do not have the true robust value function as in Eqn. (9). We start with a randomly initialized value function and expect it to gradually converge to the robust value over training. Here, we give some theoretical analysis on the convergence of this process. Let $P_n$ denote the estimated worst transition kernel at iteration $n$ and $v_{P_n}^\pi$ denote the (non-robust) value function for the transition kernel $P_n$. We are interested in the convergence of the following updates:

$$P_{n+1}(s'|s, a) = \bar{P}(s'|s, a)e^{-\frac{v_{P_n}^\pi(s') - \omega_n}{\kappa_n}}. \tag{14}$$

$\omega_n$ and $\kappa_n$ are associated with the worst case transition kernel when the target function is $v_{P_n}^\pi$. For clarity, we omit their subscript $sa$, even though they depend on $\beta_{sa}$. The following theorem shows that the value converges to the robust value and the estimated kernel converges to a worst kernel $P_\mathcal{P}^\pi$.

**Theorem 3.5.** *For the updating process in Eqn. (14), we have*

$$\|v_{P_n}^\pi - v_\mathcal{P}^\pi\|_\infty \leq \gamma^n \|v_{\bar{P}}^\pi - v_\mathcal{P}^\pi\|_\infty. \tag{15}$$

Now using the robust value function, a worst kernel $P_\mathcal{P}^\pi$ can be computed as $P^\pi(\cdot|s, a) = \bar{P}_\mathcal{P}(\cdot|s, a)e^{-\frac{v_\mathcal{P}^\pi - \omega_{sa}}{\kappa_{sa}}}$ as in Theorem 3.2.

## 4   EXPERIMENTS

In this section, we first introduce our experimental setting in Section 4.1. Next, we evaluate our method in experiments with perturbed transition dynamics, including noise perturbation (Section 4.2) and environment parameter perturbation (Section 4.3). Finally, we conduct ablation experiments for the hyperparameter $\tau$ in Section 4.4.

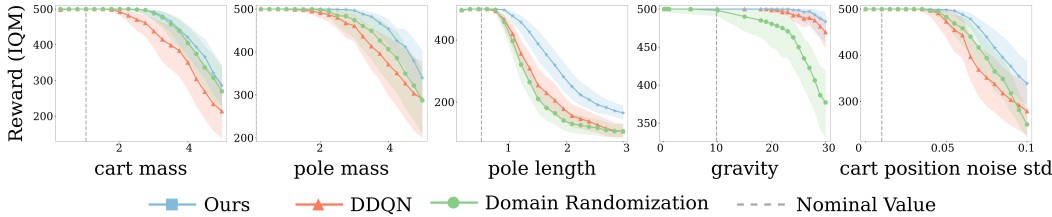

Figure 3: Evaluation results on Cartpole with perturbations

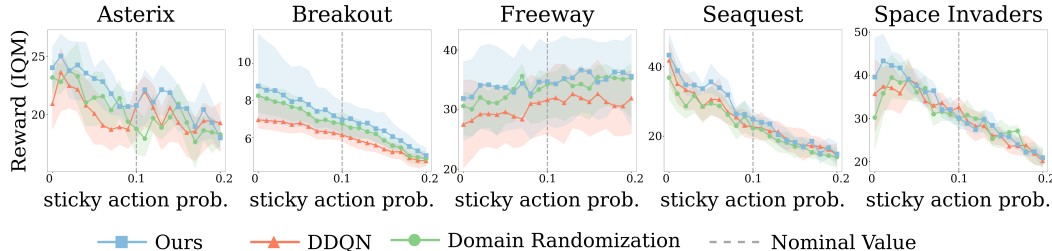

Figure 4: Evaluation results on MinAtar environments with noise perturbations.

## 4.1 SETTING

To evaluate the effectiveness of our method in learning robust policies, we conduct experiments that train the agent online under nominal dynamics and test its performance under perturbed dynamics. We consider three high-dimensional domains including both discrete and continuous control tasks, to demonstrate our algorithm can be "plugged and played" with any RL method. Specifically, we experiment on Cartpole - a classic control environment from OpenAI Gym (Brockman et al., 2016), 5 video games from the MinAtar benchmark (Young & Tian, 2019), and 4 continuous control tasks from DeepMind Control Suite (Tunyasuvunakool et al., 2020). For the baseline RL algorithm, we use Double DQN (van Hasselt et al., 2016) for classic control and MinAtar environments, and SAC (Haarnoja et al., 2018a) for continuous control environments.

As most existing methods in RMDPs literature do not scale well (see discussions in Section 5), we do not have "apple-to-apple" comparisons to those methods. So we consider another commonly-used robust RL approach as a reference: domain randomization (Tobin et al., 2017), and conduct the same set of experiments. Domain randomization trains the agent under diverse scenarios by perturbing the parameter of interest during training, such that the trained agent can be robust to similar perturbations during testing. It is worth noting that domain randomization has an edge on our method, since it has access to different perturbed parameters during training, while our method is completely oblivious to those parameters during training.

To obtain stable results, we run each experiment with multiple random seeds, and report the interquartile mean (IQM) and 95% stratified bootstrap confidence intervals (CIs) as recommended by (Agarwal et al., 2021). More details about environments, implementations, training and evaluation can be found in the appendix.

## 4.2 NOISE PERTURBATION

In this subsection, we evaluate our method in scenarios where the perturbations on the transition dynamic are implemented as noise perturbations. Specifically, we consider stochastic nominal kernels in which the stochasticity is controlled by some (observation or action) noises. The agent is trained under a fixed noise (*i.e.*, the nominal kernel) and tested with varying noises (*i.e.*, perturbed kernels).

On Cartpole, we implement the stochasticity by adding Gaussian noise to the state after applying the original deterministic dynamics of the environments, *i.e.*, $\tilde{s}_{t+1} = s_{t+1} + \epsilon$ where $\epsilon \sim \mathcal{N}(0, \sigma)$. Then $\tilde{s}_{t+1}$ is considered as the next state output from the stochastic nominal kernel. The noise scale $\sigma$ is fixed during training and varied during testing. The agent's test performance across different

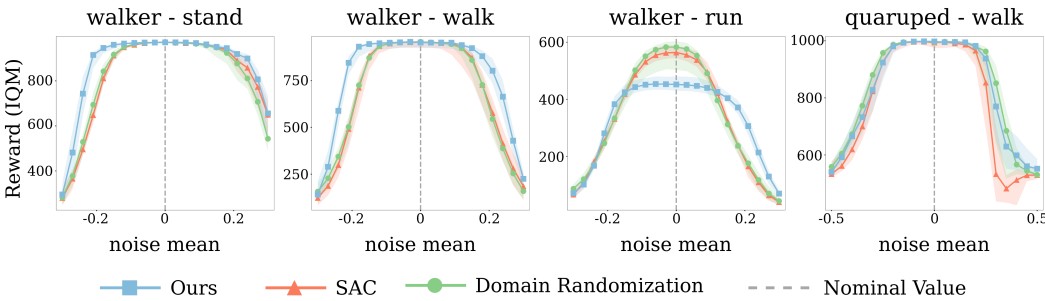

Figure 5: Evaluation results on DeepMind Control environments with noise perturbations.

perturbed values is depicted in the rightmost plot in Figure 3. When the noise scale deviates from the nominal value, EWoK achieves better performance than the baseline DDQN.

Apart from the classic control tasks, we evaluate the performance of our method in discrete control on the more challenging MinAtar environments. Here we take advantage of the existing sticky action and use it as the source of stochasticity. The sticky action probability is fixed at some value during training and perturbed during testing. As the results in Figure 4 show that EWoK yields better performance than the baseline. Sometimes the agent's performance does not follow a decreasing trend when the perturbation parameter deviates from the nominal value in one direction. This might be because the perturbation parameter has an asymmetric effect on the learning.

Next, we evaluate our method on the continuous control tasks in the DeepMind Control Suite. The stochasticity is implemented by adding Gaussian noise to the action since directly adding noise to the state might lead to an invalid physical state. During testing, we perturb the mean of the Gaussian noise. Figure 5 shows the agent's performance across different perturbed values. We can see that EWoK suffers less performance degradation as the noise mean deviates from zero (the nominal value), clearly outperforming the baseline SAC. In the walker-run task, EWoK achieves lower reward under the nominal dynamic but performs better under perturbed ones, which indicates a trade-off between the performance under the nominal kernel and robustness under perturbations.

### 4.3 PERTURBING ENVIRONMENT PARAMETERS

To further validate the effectiveness of our method, we consider a more realistic scenario where some physical/logical parameters in the environment (*e.g.*, pole length in Cartpole) are perturbed. Similarly, the agent is trained with a fixed parameter, and tested under perturbed parameters.

For Cartpole, we perturb cart mass, pole mass, pole length, and gravity. Figure 3 summarizes the testing results of the agents trained under the nominal dynamics. Again, EWoK achieves better performance than the baseline DDQN when the environment parameters deviate from the nominal value.

For DeepMind control tasks, we implement the perturbations on the environment parameters using the Real-World Reinforcement Learning Suite (Dulac-Arnold et al., 2020). Specifically, we perturb joint damping, thigh length, and torso length in walker tasks, and perturb joint damping, shin length, and torso density for quadruped tasks. As shown in Figure 6, EWoK generally works better than the baseline under model mismatch, improving the robustness of the learned policy. Similar to our observations in the previous section, the walker-run task emphasizes the inherent trade-off of solving RMDPs: optimizing the worst-case scenario can lead to suboptimal performance under the nominal model. While the performance improvement is less obvious in the quadruped-walk task, the results of our method have lower variance than the baseline.

### 4.4 ABLATION STUDIES

In this subsection, we conduct ablation experiments to investigate the effects of our hyperparameters on the performance. Recall that $\kappa$ controls the skewness of the distribution for resampling, while $N$ controls the number of next-state samples. Intuitively, when we decrease $\kappa$, we are essentially con-

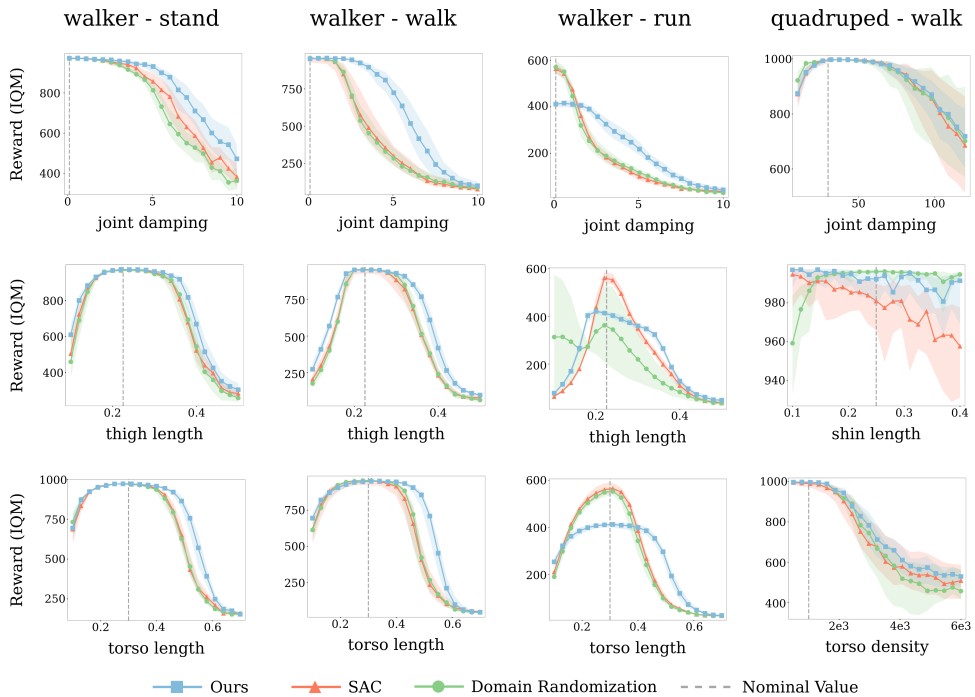

Figure 6: Evaluation results on DeepMind Control tasks with perturbed environment parameters.

sidering a higher level of robustness. If $\kappa$ is very small, then with a high probability the environment dynamic will transit to the "worst" state (*i.e.*, one with the lowest value). In addition, by increasing $N$ we effectively improve our empirical estimation of the nominal kernel's next state distribution, which should improve the worst kernel estimation.

We experiment on the DeepMind Control tasks under noise perturbation setting, using different $\kappa$ and $N$ when we train the agent. For clarity, we plot the performance difference between our method and the baseline instead of the absolute performance and defer the original results with CIs (shaded areas) to the appendix. Figure 7 shows the results of changing the values of $\kappa$. In the walker domain, decreasing $\kappa$ makes our algorithm perform better in perturbed environments, which aligns with our expectations. Figure 8 shows the results of changing the values of $N$. We can see that a small sample size will result in limited performance gain compared to the baseline, but increasing the sample size may not bring monotonic improvements. In addition, more samples will incur longer simulation time in each environment step. In our experiments, we observed minimal impact on walk-clock time, due to fast simulation. In practical scenarios where sampling next states could be slow, however, we need to take this factor into consideration. Nonetheless, we believe should not significantly increase simulation time to a prohibitive extent.

It is worth mentioning that the influence of $\kappa$ has a dependency on the environment. Decreasing it too much can lead to too conservative policies and may not always work well. For example, on the quadruped domain, using a small $\kappa$ does not yield the best performance. In addition, we observe the robustness-performance trade-off in the walker-run task once again. While large $\kappa$ achieves high performance under the nominal kernel, it significantly underperforms when the kernel is perturbed.

## 5 RELATED WORKS

Early works in RMDPs lay the theoretical foundations for solving RMDPs with robust dynamic programming (Wiesemann et al., 2013; Iyengar, 2005; Nilim & El Ghaoui, 2005; Kaufman & Schaefer, 2013; Bagnell et al., 2001). Recent works attempt to reduce the time complexity for certain uncertainty sets, such as $L_1$ uncertainty (Ho et al., 2018; 2021) and KL uncertainty (Grand-Clément & Kroer, 2021). However, they all require full knowledge of the nominal model.

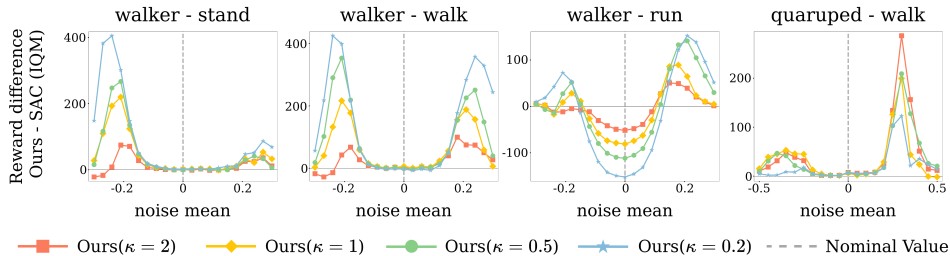

Figure 7: Evaluation results on DeepMind Control tasks with noise perturbations for different $\kappa$.

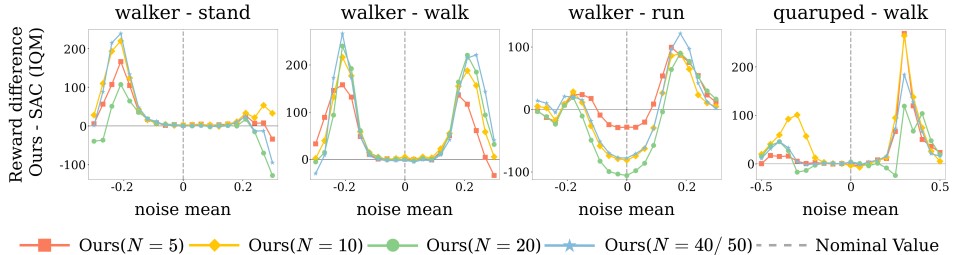

Figure 8: Evaluation results on DeepMind Control tasks with noise perturbations for different $N$.

One line of work aims to design methods that can be applied in the online robust RL setting where we do not have full knowledge about the transition model. Derman et al. (2021) define new regularized robust Bellman operators that suggest a possible online sample-based method. However, the contraction of the Bellman operators implicitly assumes that the state space can not be very large. On regularizing the learning process, Kumar et al. (2022a; 2023) introduce Q-learning and policy gradient methods for $L_p$ uncertainty sets, but do not experimentally evaluate their methods with experiments. Another type of uncertainty is the R-contamination, for which previous works have derived a robust Q-learning algorithm (Wang & Zou, 2021) and a regularized policy gradient algorithm (Wang & Zou, 2022a). R-contamination uncertainty assumes that the adversary can take the agent to any state, which is too conservative in practice. In addition, all of those methods are tied to a particular type of RL algorithm. Our work, however, aims to tackle the problem from a different perspective by approximating a worst kernel and can adopt any non-robust RL algorithm to learn an optimal robust policy. A recent work (Wang et al., 2023) has shown that the worst kernel can be computed using gradient descent, but their method takes more iterations to converge ($O(\frac{S^3 A}{(1-\gamma)^6 \epsilon^2})$) compared to ours ($O(\log \frac{1}{\epsilon})$).

Our work is also closely related to (Kumar et al., 2023), which characterizes the worst kernel for $L_p$ uncertainty set. Different from their work, we propose to approximately simulate this worst kernel, opening a new paradigm for learning robust policies in RMDPs. In addition, our work focuses on the KL-regularized uncertainty, a setting more realistic than the $L_p$ case. Under $L_p$ uncertainty, we essentially consider perturbations that might take the agent to any state but real perturbations are often local. In comparison, perturbations under KL-regularized uncertainty only focus on states where the nominal kernel has a non-zero transition probability.

## 6    CONCLUSIONS AND DISCUSSIONS

In this paper, we introduce an approach that tackles the RMDPs problem from a new perspective, by approximately simulating a worst transition kernel while leaving the RL part untouched. The highlight of our method is that it can be applied on top of existing non-robust deep RL algorithms to learn robust policies, exhibiting attractive scalability to high-dimensional domains. We believe this new perspective will offer some insights for future works on RMDPs. One limitation of our work is that we require the ability to sample next states from the transition model multiple times. In future works, we will study how to combine our method with a learned transition model where sampling the next states would not be a problem.

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

# A PROOF

## A.1 PROOF OF THEOREM 3.2

Recall that the worst values are defined as

$$P_{\mathcal{P}}^{\pi} \in \arg\min_{P \in \mathcal{P}} J_P^{\pi}$$

for any general uncertainty set $\mathcal{P}$. Further, for $sa$-rectangular uncertainty set $\mathcal{P} = \times_{s \in \mathcal{S}, a \in \mathcal{A}} \mathcal{P}_{sa}$, the robust value function exists, that is, the following is well defined (Nilim & El Ghaoui, 2005; Iyengar, 2005)

$$v_{\mathcal{P}}^{\pi} = \min_{P \in \mathcal{P}} v_P^{\pi}.$$

This implies,

$$v_{\mathcal{P}}^{\pi} = \left( I - \gamma (P_{\mathcal{P}}^{\pi})^{\pi} \right)^{-1} R^{\pi}$$

is the fixed point of robust Bellman operator $\mathcal{T}_{\mathcal{P}}^{\pi}$ (Nilim & El Ghaoui, 2005; Iyengar, 2005), defined as

$$\mathcal{T}_{\mathcal{P}}^{\pi} v := \min_{P \in \mathcal{P}} \mathcal{T}_P^{\pi} v.$$

**Proposition A.1.** *The worst values can be computed from the robust value function. That is*

$$\arg\min_{P \in \mathcal{P}} \mathcal{T}_P^{\pi} v_{\mathcal{P}}^{\pi} \subseteq \arg\min_{P \in \mathcal{P}} v_P^{\pi} \subseteq \arg\min_{P \in \mathcal{P}} J_P^{\pi}.$$

*Proof.* Let

$$P^* \in \arg\min_{P \in \mathcal{P}} \mathcal{T}_P^{\pi} v_{\mathcal{P}}^{\pi}.$$

Now, from the fixed point of robust Bellman operator, we have

$$
\begin{aligned}
v_{\mathcal{P}}^{\pi} = \mathcal{T}_{\mathcal{P}}^{\pi} v_{\mathcal{P}}^{\pi} &= \min_{P \in \mathcal{P}} \mathcal{T}_P^{\pi} v_{\mathcal{P}}^{\pi}, \\
&= \mathcal{T}_{P^*}^{\pi} v_{\mathcal{P}}^{\pi}, \qquad \text{(by construction)}, \\
&= R^{\pi} + \gamma (P^*)^{\pi} v_{\mathcal{P}}^{\pi}, \qquad \text{(by definition)}.
\end{aligned}
$$

The above implies,

$$v_{\mathcal{P}}^{\pi} = \left( I - \gamma (P^*)^{\pi} \right)^{-1} R^{\pi}.$$

This implies,

$$P^* \in \arg\min_{P \in \mathcal{P}} v_P^{\pi}.$$

The last inclusion is trivial, that is, every minimizer of value function is a minimizer of robust return. $\qquad\square$

**Theorem 3.2.** *For a KL uncertainty set $\mathcal{P}$, a worst kernel is related to the nominal kernel through:*

$$P_{\mathcal{P}}^{\pi}(s'|s, a) = \bar{P}^{\pi}(s'|s, a) e^{-\delta^{\pi}(s')}, \tag{8}$$

*where $\delta^{\pi}$ is of the form*

$$\delta^{\pi}(s') = \frac{v_{\mathcal{P}}^{\pi}(s') - \omega_{sa}}{\kappa_{sa}}, \tag{9}$$

*and satisfies*

$$\sum_{s'} \bar{P}^{\pi}(s'|s, a) e^{-\delta^{\pi}(s')} = 1, \quad \sum_{s'} \bar{P}^{\pi}(s'|s, a) e^{-\delta^{\pi}(s')} (-\delta^{\pi}(s')) = \beta_{sa}. \tag{10}$$

*Proof.* Recall Definition 3.1

$$P_{\mathcal{P}}^{\pi} \in \arg\min_{P \in \mathcal{P}} J_P^{\pi}. \tag{16}$$

From Proposition A.1, for sa-rectangular uncertainty set $\mathcal{P}$, a worst kernel can be computed using robust value function as

$$P_{\mathcal{P}}^{\pi} \in \arg\min_{P \in \mathcal{P}} \mathcal{T}_P^{\pi} v_{\mathcal{P}}^{\pi}.$$

Recall, our KL-constrained uncertainty $\mathcal{P}$ is defined as

$$\mathcal{P} := \{P \mid P \in (\Delta_{\mathcal{S}})^{\mathcal{S} \times \mathcal{A}}, D_{KL}(\bar{P}_{s,a}, P_{sa}) \leq \beta_{sa}, \forall s, a\}.$$

where $D_{KL}$ is KL norm that is defined as

$$D_{KL}(P, Q) = \sum_s P(s) \log\left(\frac{P(s)}{Q(s)}\right).$$

Using Proposition A.1 and definition of uncertainty set $\mathcal{P}$, the worst kernel can be extracted as

$$P_{\mathcal{P}}^{\pi}(\cdot|s, a) \in \arg\min_{D_{KL}(p, P_0(\cdot|s,a)) \leq \beta_{sa}, \sum_s p(s) = 1, p \succeq 0} \langle p, v_{\mathcal{U}}^{\pi} \rangle.$$

Using the Lemma A.2, we get the desired solution. $\qquad\square$

**Lemma A.2.** *For $q \in \Delta_{\mathcal{S}}, v \in \mathbb{R}^{\mathcal{S}}, \beta \geq 0$, a solution to*

$$\min_{p \ln(\frac{p}{q}) \leq \beta, 1^T p = 1, p \succeq 0} \langle p, v \rangle.$$

*is given by*

$$p = q e^{-\frac{v-\omega}{\lambda}},$$

*where*

$$p \log(\frac{p}{q}) = \left\langle q e^{-\frac{v-\omega}{\lambda}}, \frac{v-\omega}{\lambda} \right\rangle = -\beta$$

*and*

$$\sum_s q(s) e^{-\frac{v(s)-\omega}{\lambda}} = 1.$$

*Proof.* We have the following optimization problem,

$$\min_{p \ln(\frac{p}{q}) \leq \beta, 1^T p = 1, p \succeq 0} \langle p, v \rangle. \tag{17}$$

We ignore the constraint $p \succeq 0$ for the moment (as we see later, this constrained is automatically satisfied), and focus on

$$\min_{p \ln(\frac{p}{q}) \leq \beta, 1^T p = 1} \langle p, v \rangle. \tag{18}$$

We define Lagrange multiplier as

$$L(p, \lambda, \mu) = \langle p, v \rangle + \lambda \left( p \ln(\frac{p}{q}) - \beta \right) + \mu \left( 1^T p - 1 \right).$$

We now put the stationarity condition:

$$\frac{\partial L}{\partial p} = v + \lambda \left( \ln(\frac{p}{q}) + 1 \right) + \mu 1 = 0$$
$$\implies p = q e^{-1} e^{-\frac{v+\mu}{\lambda}}.$$

With appropriate change of variable $\mu \to \omega$, we have

$$p = q e^{-\frac{v-\omega}{\lambda}}.$$

We have to find the constants $\omega$ and $\lambda$, using the constraints

$$p\log(\frac{p}{q}) = \left\langle qe^{-\frac{v-\omega}{\lambda}}, \frac{v-\omega}{\lambda} \right\rangle = -\beta$$

and

$$\sum_s p(s) = \sum_s q(s)e^{-\frac{v(s)-\omega}{\lambda}} = 1.$$

We further note that the constraint $1 \geq p(s) \geq 0$ is automatically satisfied as

$$p(s) = q(s)e^{-\frac{v(s)-\omega}{\lambda}} \geq 0$$

and $\sum_s p(s) = 1$, ensures $p(s) \leq 1 \quad \forall s$. $\qquad\square$

## A.2 PROOF OF PROPOSITION 3.3 AND 3.4

**Proposition A.3.** $\omega_{sa}$ *can be upper-bounded as follows,*

$$\omega_{sa} \leq \langle \bar{P}(\cdot|s,a), v_{\mathcal{P}}^\pi \rangle, \qquad \forall s \in \mathcal{S}, a \in \mathcal{A}.$$

*Proof.* From the constraint in Theorem 3.2, we have

$$\sum_{s'} \bar{P}(s'|s,a)e^{-\frac{v_{\mathcal{P}}^\pi(s')-\omega_{sa}}{\kappa_{sa}}} = 1 \tag{19}$$

$$\implies e^{-\sum_{s'} \bar{P}(s'|s,a)\frac{v_{\mathcal{P}}^\pi(s')-\omega_{sa}}{\kappa_{sa}}} \leq 1 \qquad \text{(using Jenson's inequality)} \tag{20}$$

$$\implies e^{-\sum_{s'} \bar{P}(s'|s,a)\frac{v_{\mathcal{P}}^\pi(s')}{\kappa_{sa}}} e^{\frac{\omega_{sa}}{\kappa_{sa}}} \leq 1 \tag{21}$$

$$\implies \frac{\omega_{sa}}{\kappa_{sa}} \leq \sum_{s'} \bar{P}(s'|s,a)\frac{v_{\mathcal{P}}^\pi(s')}{\kappa_{sa}} \tag{22}$$

$$\implies \omega_{sa} \leq \sum_{s'} \bar{P}(s'|s,a)v_{\mathcal{P}}^\pi(s'). \qquad\qquad\square$$

**Proposition 3.3.** $\omega_{sa}$, $\kappa_{sa}$ *and* $\beta_{sa}$ *satisfy*

$$\omega_{sa} = \langle P_{\mathcal{P}}^\pi(\cdot|s,a), v_{\mathcal{P}}^\pi \rangle + \beta_{sa}\kappa_{sa}, \tag{11}$$

*Proof.* From the constraint in Theorem 3.2, we have

$$\sum_{s'} \bar{P}^\pi(s'|s,a)e^{-\delta^\pi(s')}(-\delta^\pi(s')) = \beta_{sa} \tag{23}$$

$$\implies \sum_{s'} P_{\mathcal{P}}^\pi(s'|s,a)\frac{v_{\mathcal{P}}^\pi(s')-\omega_{sa}}{\kappa_{sa}} = \beta_{sa} \tag{24}$$

$$\implies \sum_{s'} P_{\mathcal{P}}^\pi(s'|s,a)v_{\mathcal{P}}^\pi(s') = -\beta_{sa}\kappa_{sa} + \omega_{sa}. \qquad\square$$

**Proposition 3.4.** $\omega_{sa}$ *can be bounded as follows,*

$$\langle P_{\mathcal{P}}^\pi(\cdot|s,a), v_{\mathcal{P}}^\pi \rangle \leq \omega_{sa} \leq \langle \bar{P}^\pi(\cdot|s,a), v_{\mathcal{P}}^\pi \rangle. \tag{13}$$

*Proof.* The lower bound is direct from Proposition 3.3, as $\beta$ and $\kappa$ are positive quantities by definition. The upper bound comes from Proposition A.3. $\qquad\square$

## A.3 PROOF OF THEOREM 3.5

Given a policy $\pi$, let $P_{n+1}$ be the updated kernel:

$$P_{n+1} = \arg\min_{P \in \mathcal{P}} T_P^\pi v_{P_n}^\pi. \tag{25}$$

We continue to prove the following lemmas.

**Lemma A.4.** *The kernel update process produces monotonically decreasing value functions:*

$$v^\pi_{P_n} \succeq v^\pi_{P_{n+1}}, \quad \forall n = 1, 2, \cdots. \tag{26}$$

*Proof.* Recall that $v^\pi_{P_n} = T^\pi_{P_n} v^\pi_{P_n} = R^\pi + \gamma P^\pi_n v^\pi_{P_n}$. Since we have

$$P_{n+1} = \arg\min_{P \in \mathcal{P}} [R^\pi + \gamma P^\pi v^\pi_{P_n}], \tag{27}$$

we can obtain

$$R^\pi + \gamma P^\pi_n v^\pi_{P_n} \geq \min_{P \in \mathcal{P}} [R^\pi + \gamma P^\pi v^\pi_{P_n}]$$
$$\Rightarrow \qquad v^\pi_{P_n} \geq R^\pi + \gamma P^\pi_{n+1} v^\pi_{P_n}$$
$$\Rightarrow \qquad (I - \gamma P^\pi_{n+1}) v^\pi_{P_n} \geq R^\pi$$
$$\Rightarrow \qquad v^\pi_{P_n} \geq (I - \gamma P^\pi_{n+1})^{-1} R^\pi = v^\pi_{P_{n+1}}. \qquad \square$$

**Lemma A.5.** *The robust bellman operators are monotonic functions, that is:*

$$v \leq u \implies T^\pi_\mathcal{P} v \leq T^\pi_\mathcal{P} u$$

*Proof.* Since $v \leq u$, and the fact that $P$ has only non-negative entries, we know that:

$$R^\pi + \gamma P^\pi v \leq R^\pi + \gamma P^\pi u, \quad \forall P \in \mathcal{P}$$
$$\Rightarrow \qquad \min_{P \in \mathcal{P}} (R^\pi + \gamma P^\pi v) \leq \min_{P \in \mathcal{P}} (R^\pi + \gamma P^\pi u)$$
$$\Rightarrow \qquad T^\pi_\mathcal{P} v \leq T^\pi_\mathcal{P} u$$

$$\square$$

**Theorem 3.5.** *For the updating process in Eqn. (14), we have*

$$\|v^\pi_{P_n} - v^\pi_\mathcal{P}\|_\infty \leq \gamma^n \|v^\pi_P - v^\pi_\mathcal{P}\|_\infty. \tag{15}$$

*Proof.* We prove it by showing that:

$$\|v^\pi_{P_{n+1}} - v^\pi_\mathcal{P}\|_\infty \leq \gamma \|v^\pi_{P_n} - v^\pi_\mathcal{P}\|_\infty, \quad \forall n. \tag{28}$$

First, by optimality, we have

$$v^\pi_{P_{n+1}} - v^\pi_\mathcal{P} \geq 0. \tag{29}$$

Now we can focus only on the upper bound:

$$v^\pi_{P_{n+1}} - v^\pi_\mathcal{P} = T^\pi_{P_{n+1}} v^\pi_{P_{n+1}} - T^\pi_\mathcal{P} v^\pi_\mathcal{P}$$
$$\leq T^\pi_{P_{n+1}} v^\pi_{P_n} - T^\pi_\mathcal{P} v^\pi_\mathcal{P} \qquad \text{(Lemma A.4 and A.5)}$$
$$= \min_{P \in \mathcal{P}} T^\pi_P v^\pi_{P_n} - T^\pi_\mathcal{P} v^\pi_\mathcal{P}$$
$$= T^\pi_\mathcal{P} v^\pi_{P_n} - T^\pi_\mathcal{P} v^\pi_\mathcal{P}$$
$$\leq \gamma \|v^\pi_{P_n} - v^\pi_\mathcal{P}\|_\infty \qquad \text{($T^\pi_\mathcal{P}$ is a $\gamma$-contraction operator).}$$

Putting it together, we have

$$\|v^\pi_{P_{n+1}} - v^\pi_\mathcal{P}\|_\infty \leq \gamma \|v^\pi_{P_n} - v^\pi_\mathcal{P}\|_\infty.$$

The desired result is proved by applying the above result iteratively. $\qquad \square$

## B    Experiment details

### B.1    Environments

#### B.1.1    Classic control tasks

Cartpole [1] is one of the classic control tasks in OpenAI Gym (Brockman et al., 2016). The task is to balance a pendulum on a moving cart, by moving the cart either left or right. The state consists of the location and velocity of the cart, as well as the angle and angular velocity of the pendulum. To make the transition dynamic stochastic, we add Gaussian noises to the cart position or the pole angle. The detailed configurations for the nominal values and the perturbation ranges are summarized in Table 1.

Table 1: Perturbation configurations for Cartpole environment.

|  | Parameter | Nominal value | Perturbation range |
|---|---|---|---|
| Noise perturbation | Cart position noise (std) | 0.01 | $[0, 0.1]$ |
|  | Pole angle noise (std) | 0.01 | $[0, 0.05]$ |
| Env. param. pertubration | Pole mass | 0.1 | $[0.15, 3.0]$ |
|  | Pole length | 0.5 | $[0.25, 5.0]$ |
|  | Cart mass | 1 | $[0.25, 5.0]$ |
|  | Gravity | 9.8 | $[0.1, 30]$ |

#### B.1.2    MinAtar

The MinAtar benchmark[2] is a simplification of the widely-used Atari benchmark (Bellemare et al., 2013), which eliminates the representation complexity while preserving the game mechanism. MinAtar consists of 5 games: `Asterix`, `Breakout`, `Freeway`, `Seaquest` and `SpaceInvaders`. The observation is a $10 \times 10 \times n$ grid image, where each channel corresponds to a game-specific object. We use the minimal action space for each game. As mentioned in the main text, the stochasticity of the transition dynamic comes from *sticky actions*. That is, at each step, the agent would repeat the previous action with some probability instead of executing the chosen action. The nominal value for sticky action probability is $0.1$ and the perturbation range is $[0.0, 0.2]$.

#### B.1.3    DeepMind Control Suite

The DeepMind Control Suite (Tunyasuvunakool et al., 2020) is a set of continuous control tasks powered by the MuJoCo physics engine (Todorov et al., 2012). It is widely used to benchmark reinforcement learning agents. As mentioned in the main text, we consider 4 tasks in our paper: `walker-stand`, `walker-walk`, `walker-run`, `quadruped-walk`. For `walker` tasks, the observations are 24-dimensional vectors and the actions are 6-dimensional vectors. For `quadruped`, the observations are 78-dimensional vectors and the actions are 12-dimensional vectors. For noise perturbation, we fix the standard deviation of the Gaussian noise to 0.2 for `walker` and 0.1 for `quadruped`. The nominal value and the perturbation range are summarized in Table 2 and Table 3.

### B.2    Training and evaluation

For both our method and the baseline, we first train the agent under the nominal environment, and then for each perturbed environment during testing, we calculate the average reward from 30 episodes. We repeat this process with 40 random seeds in the classic control environments and 10 seeds in MinAtar and DeepMind Control environments. Following the recommended practice in (Agarwal et al., 2021), we report the Interquartile Mean (IQM) and the $95\%$ stratified bootstrap confidence intervals (CIs), using The IQM metric is measured by discarding the top and bottom $25\%$

---

[1] https://gymnasium.farama.org/environments/classic_control/cart_pole/
[2] https://github.com/kenjyoung/MinAtar

Table 2: Perturbation configurations for walker tasks.

|  | PARAMETER | NOMINAL VALUE | PERTURBATION RANGE |
|---|---|---|---|
| NOISE PERTUBRATION | action noise (mean) | 0.0 | $[-0.3, 0.3]$ |
| ENV. PARAM. PERTUBRATION | thigh length | 0.225 | $[0.1, 0.5]$ |
|  | torso length | 0.3 | $[0.1, 0.7]$ |
|  | joint damping | 0.1 | $[0.1, 10]$ |
|  | contact friction | 0.7 | $[0.01, 0.7]$ |

Table 3: Perturbation configurations for quadruped tasks.

|  | PARAMETER | NOMINAL VALUE | PERTURBATION RANGE |
|---|---|---|---|
| NOISE PERTUBRATION | action noise (mean) | 0.0 | $[-0.5, 0.5]$ |
| ENV. PARAM. PERTUBRATION | shin length | 0.25 | $[0.1, 0.4]$ |
|  | torso density | 1000 | $[500, 6000]$ |
|  | joint damping | 30 | $[10, 120]$ |
|  | contact friction | 1.5 | $[0.1, 2.5]$ |

of the results, and averaging across the remaining middle 50%. IQM has the benefit of being more robust to outliers than a regular mean, and being a better estimator of the overall performance than the median. We use the rliable library[3] to calculate IQM and CIs.

As mentioned earlier, we use Double-DQN (van Hasselt et al., 2016) as the vanilla non-robust RL algorithm for environments with discrete action spaces. Specifically, we follow the implementation in Stable-Baselines3 (Raffin et al., 2021). For the classic control environments, we use the DDQN's hyperparameters suggested in RL Baselines3 Zoo (Raffin, 2020), and for the MinAtar environments, we use the hyperparameters described by (Young & Tian, 2019). For the Cartpole environment, we use a two-layer MLP neural network with 256 hidden units per layer. For the MinAtar environments, we use a CNN consisting of a single convolutional layer (16 output channels, $3 \times 3$ kernel, stride= 1, and padding= 0) and another fully connected layer with 128 hidden units. The detailed configurations are summarized in Table 4.

Table 4: Hyperparameters for DDQN used in the experiments.

| PARAMETER | CARTPOLE | MINATAR |
|---|---|---|
| batch size | 64 | 32 |
| buffer size | 100000 | 100000 |
| exploration final epsilon | 0.04 | 0.01 |
| exploration fraction | 0.16 | 0.1 |
| gamma | 0.99 | 0.99 |
| gradient steps | 128 | 1 |
| learning rate | 0.0023 | 0.00025 |
| learning starts | 1000 | 5000 |
| target update interval | 10 | 1000 |
| train frequency | 256 | 4 |
| total time-steps | 50000 | 5000000 |

For environments with continuous action spaces, we choose the SAC algorithm (Haarnoja et al., 2018b) as the vanilla non-robust RL algorithm, and follow the implementations and hyperparameter

---

[3]https://github.com/google-research/rliable

choices in (Yarats & Kostrikov, 2020). Both the actor and critic use a two-layer MLP neural network with 1024 hidden units per layer. Table 5 lists the hyperparameters.

Table 5: Hyperparameters for SAC used in the experiments.

| PARAMETER | VALUE |
|---|---|
| Total steps | 1e6 |
| Warmup steps | 5000 |
| Replay size | 1e6 |
| Batch size | 1024 |
| Discount factor $\gamma$ | 0.99 |
| Optimizer | Adam (Kingma & Ba, 2015) |
| Learning rate | 1e-4 |
| Target smoothing coefficient | 0.005 |
| Target update interval | 2 |
| Initial temperature | 0.1 |
| Learnable temperature | Yes |

The configurations for the sample size $N$ and the robustness parameter $\kappa$ used in our experiments are summarized in Table 6.

Table 6: Hyperparameters specific to our method used in the experiments.

| | ENVIRONMENT | SAMPLE SIZE $N$ | ROBUSTNESS PARAMETER $\kappa$ |
|---|---|---|---|
| CLASSIC CONTROL | Cartpole | 15 | 1 |
| MINATAR | Asterix | 5 | 5 |
| | Breakout | 50 | 5 |
| | Freeway | 5 | 1 |
| | Seaquest | 10 | 10 |
| | SpaceInvaders | 5 | 0.5 |
| DEEPMIND CONTROL SUITE | walker-stand | 10 | 0.2 |
| | walker-walk | 10 | 0.2 |
| | walker-run | 10 | 0.2 |
| | quadruped-walk | 40 | 0.5 |

### B.3 COMPUTATIONAL RESOURCES AND COSTS

We used the following resources in our experiments:

- **CPU:** AMD EPYC 7742 64-Core Processor
- **GPU:** NVIDIA GeForce RTX 2080 Ti

Table 7 lists the training time.

Table 7: Training time per run of our experiments on a single GPU.

|  | ENVIRONMENT | BASELINE | OURS |
|---|---|---|---|
| CLASSIC CONTROL | Cartpole | $\sim 4$ minutes | $\sim 5$ minutes |
| MINATAR | Asterix | $\sim 4$ hours | $\sim 5$ hours |
|  | Breakout | $\sim 3$ hours | $\sim 4$ hours |
|  | Freeway | $\sim 9.5$ hours | $\sim 12$ hours |
|  | Seaquest | $\sim 8$ hours | $\sim 4.5$ hours |
|  | SpaceInvaders | $\sim 4$ hours | $\sim 6$ hours |
| DEEPMIND CONTROL SUITE | all tasks | $\sim 5$ hours | $\sim 6$ hours |

## C ADDITIONAL RESULTS

Even though we used SAC in our main results, to further exert our claim that EWoK can be applied on any off-the-shelf non-robust RL algorithm, we repeated the continuous control tasks experiments with the TD3 (Fujimoto et al., 2018) algorithm as a baseline. The results are depicted in Figures 9 and 10.

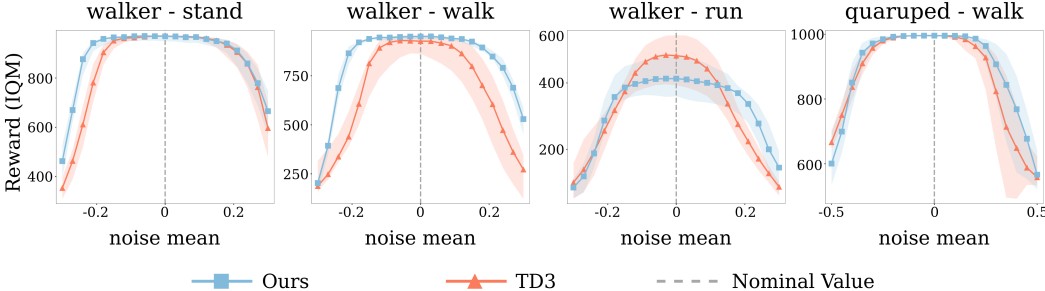

Figure 9: Evaluation results on DeepMind Control environments with noise perturbations.

In section 4.4, we show the relative performance for the ablation study on parameter $\kappa$. Here we include the absolute results in Figures 11 and 12.

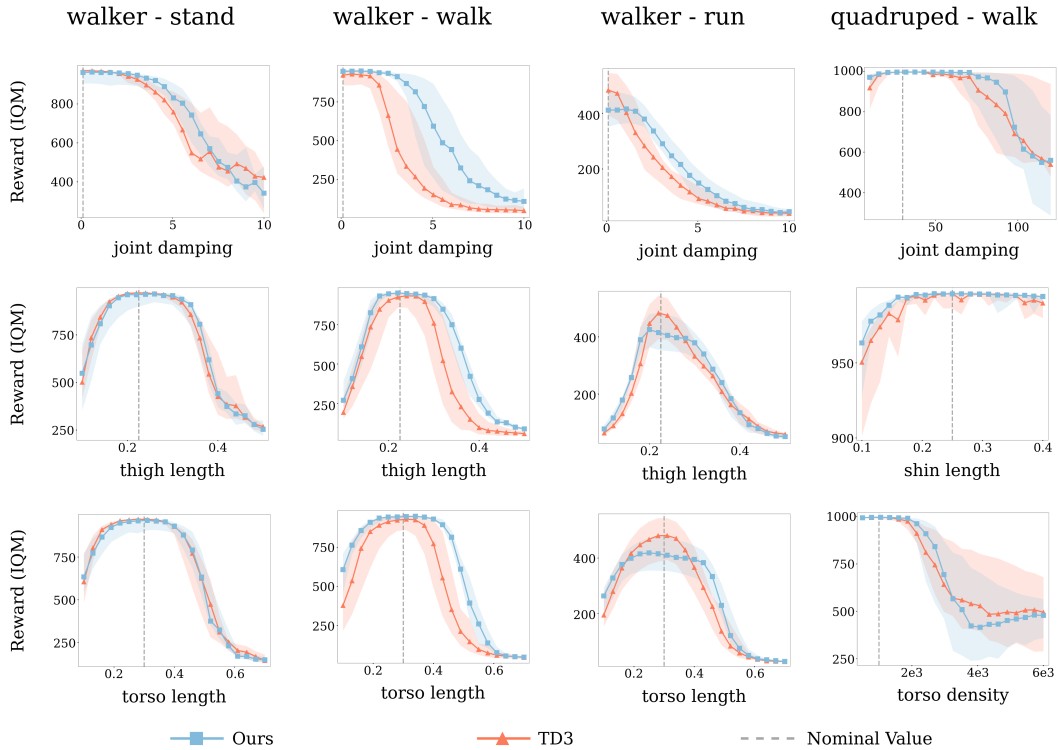

Figure 10: Evaluation results on DeepMind Control tasks with perturbed environment parameters.

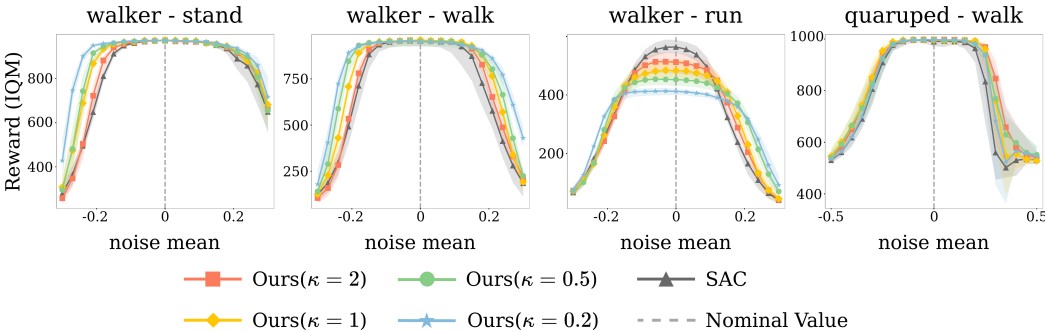

Figure 11: Evaluation results on DeepMind Control tasks with noise perturbations for different $\kappa$.

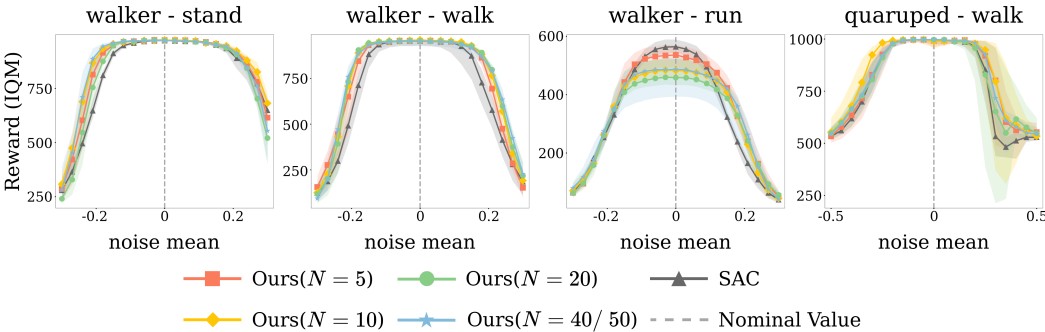

Figure 12: Evaluation results on DeepMind Control tasks with noise perturbations for different $N$.

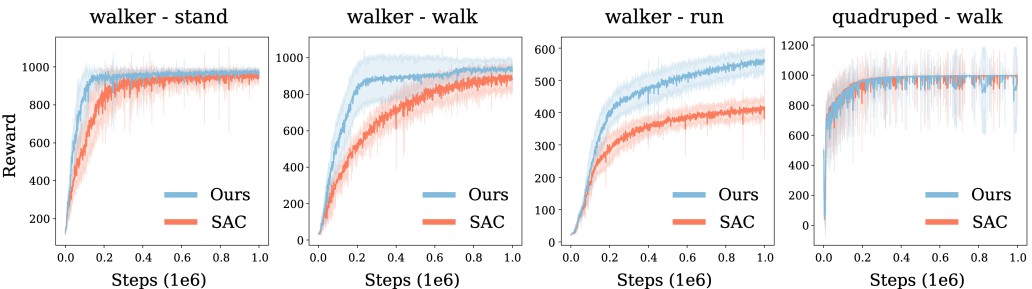

Figure 13: Training curves of experiments on DeepMind Control tasks.

