# OpenReview forum: "EWoK: Tackling Robust Markov Decision Processes via Estimating Worst Kernel"
_ICLR.cc/2024/Conference — Submitted to ICLR 2024_

### Official Review · Reviewer_rzcF · 2023-10-16

**Soundness:** 2 fair
**Presentation:** 2 fair
**Contribution:** 1 poor
**Rating:** 3
**Confidence:** 4

**Summary:**

The authors introduce a new method to learn robust policies by approximately simulating the worst-case transition probabilities. The method works for KL-divergence based sa-rectangular uncertainty set. A large set of numerical experiments is conducted.

**Strengths:**

* The authors conduct large-scale experiments.
* It is nice that we can combine the methods proposed in the paper with other (non-robust) RL algorithms.

**Weaknesses:**

* “The optimal policy”, “The worst-case transition probabilities”: some there are multiple optimal policies/worst-case kernels, I think that the authors should be more careful with the phrasing. I also don’t know what it means to “train” a transition kernel (last sentence of the paragraph after Eq. (7)), and I don’t know what is a “perfect value function” (paragraph before Eq. (14)). I list other inaccurate statements below.

* The Theoretical results are very weak. Nothing new in Appendix A.1 (proof of Theorem 3.2), it is already in [A].

* The Theoretical results are only for KL divergence.

[A] A. Nilim and L. El Ghaoui. Robust control of Markov decision processes with uncertain transition probabilities. Operations Research, 53(5):780–798, 2005.

**Questions:**

1. Can the EWOK be modified to cope with other uncertainty sets than KL-based uncertainty?

2. For completeness, can you recall how to tune $\beta_{sa}$ the uncertainty radius, or give a precise reference for this?

3. Paragraph after Eq. (7): it is now well-recognized that the minimization problem from Eq. (7) is also an MDP, see for instance Section 3 in [1] or Section 4 in [2]. Can we use gradient-based method in this adversarial MDP to learn the worst-case transition probabilities?

4. Please properly introduce the parameter $\omega_{sa}$ and $\kappa_{sa}$ in Th 3.2.

5. Th 3.5 only states that the value converges to the robust value. It does not state that “the estimated kernel converges to the worst kernel”. Also why is \pi not indexed by the iteration, since you update it at every iteration?

[1] Vineet Goyal and Julien Grand-Clement. Robust Markov decision processes: Beyond rectangularity. Mathematics of Operations Research, 2022.

[2] Chin Pang Ho, Marek Petrik, and Wolfram Wiesemann. Partial policy iteration for l1-robust
Markov decision processes. The Journal of Machine Learning Research, 22(1):12612–12657, 2021

---

> ### Author Response · Authors · 2023-11-14
> **Authors' response (Part I)**
>
> We extend our gratitude for your valuable and constructive feedback. It is truly encouraging to see the recognition of our efforts in large-scale experiments. Below, we address the comments and questions and are happy to engage in further discussions.
>
> >
>
> `Question 1 (Q1):` “The optimal policy”, “The worst-case transition probabilities”: some there are multiple optimal policies/worst-case kernels, I think that the authors should be more careful with the phrasing. I also don’t know what it means to “train” a transition kernel (last sentence of the paragraph after Eq. (7)), and I don’t know what is a “perfect value function” (paragraph before Eq. (14)). I list other inaccurate statements below.
>
> `Response 1 (R1):` Thank you for pointing out the inaccuracies in our phrasing. In the updated version, we have rectified these to 'An optimal policy' and 'A worst-case kernel'. Concerning 'train a transition kernel', our intent was to refer to the process of learning a parameterized transition model. As for 'perfect robust value function', we aimed to convey that we do not have the true robust value function beforehand in practice. These imprecise statements have been addressed in the updated version.
>
> >
>
> `Q2:` The Theoretical results are very weak. Nothing new in Appendix A.1 (proof of Theorem 3.2), it is already in [A].
>
> `R2`: We agree that Theorem 3.2 is not difficult to derive. We present its proof for the sake of completeness. Our primary contribution lies in two key aspects:
>
> * Conceptually, we show how to use Theorem 3.2 for sampling the worst kernel using the nominal kernel.
> * The core theoretical contribution is Theorem 3.5, which is the heart of our algorithm. The update rule in Equation 14 is coupled, i.e., we update the transition kernel $P_{n+1}$ using the value function w.r.t. the kernel $P_n$. Hence, showing its convergence is non-trivial, necessitating a detailed and technical proof in Appendix A.1. We believe this proof does not reside in [A] but rather constitutes our core results.
>
> Thus, we respectfully disagree with the reviewer's assessment of our theoretical contribution as weak.
>
> >
>
> `Q3:` Can the EWOK be modified to cope with other uncertainty sets than KL-based uncertainty?
>
> `R3`: Yes, we believe the high-level idea of EWoK, approximating a worst transition kernel, can be applied in other uncertainty sets than the KL-based one. The key step is to build the connection between a worst transition kernel and the nominal one, and use such connection to obtain samples approximately distributed according to the worst transition probability.
>
> >
>
> `Q4:` For completeness, can you recall how to tune $\beta_{sa}$ the uncertainty radius, or give a precise reference for this?
>
> `R4`: $\beta_{sa}$ is more of a parameter for modeling purposes. In our experiments, we do not tune $\beta_{sa}$ but set $\omega$ and $\kappa$ instead.
>
> >
>
> `Q5:` Paragraph after Eq. (7): it is now well-recognized that the minimization problem from Eq. (7) is also an MDP, see for instance Section 3 in [1] or Section 4 in [2]. Can we use gradient-based method in this adversarial MDP to learn the worst-case transition probabilities?
>
> `R5`: This is a good point. Thank you! We concur that the worst kernel can be computed using gradient descent, as described in Algorithm 2 in [A]. However, the convergene of Algorithm 2 to the worst kernel, has the iteration complexity of $O(\frac{S^3A}{(1-\gamma)^6\epsilon^2})$ (as guaranteed by their Theorem 4.4). In comparison, our method (update rule in our Eqn. 14) has the iteration complexity of $O(\log(\frac{1}{\epsilon}))$. We thank the reviewer for pointing out this direction and we have included the above discussion in the updated version.
>
> [A] Qiuhao Wang,Chin Pang Ho, Marek Petrik. Policy Gradient in Robust MDPs with Global Convergence Guarantee

---

> > ### Author Response · Authors · 2023-11-14
> > **Authors' response (Part II)**
> >
> > `Q6:` Please properly introduce the parameter $\omega_{sa}$ and $\kappa_{sa}$ in Th 3.2.
> >
> > `R6`: Thank you for the suggestion. We have revised our paper to make it more clear.
> >
> > >
> >
> > `Q7:` Th 3.5 only states that the value converges to the robust value. It does not state that “the estimated kernel converges to the worst kernel”. Also why is \pi not indexed by the iteration, since you update it at every iteration?
> >
> > `R7`: Thank you for highlighting this aspect. We note that the closeness between a kernel and a worst kernel can be measured by the difference in the value function. An example can be found in Theorem 4.4 in [A]. We call two kernels "close" if their value functions are close, because most of the time we care how a kernel minimizes the value (recall the definition of worst kernel).
> >
> > Theorem 3.5 is true for all policies $\pi$ (hence all $\pi_k$). However, we can easily incorporate the changing $\pi_k$, by utilizing multi-scale algorithm reasoning. That is, running $\pi_k$ on the slower time scale and $P_k$ on the faster time scale, which allows us to decouple both updates. However, we avoided doing so, as our method is a *meta-algorithm*. It can be combined with *any* method to update the policy. This is the core message of our paper, that we simulate the worst kernel, which can be combined with any non-robust RL method to learn the robust policy.
> >
> > [A] Qiuhao Wang,Chin Pang Ho, Marek Petrik. Policy Gradient in Robust MDPs with Global Convergence Guarantee

---

> > ### Comment · Reviewer_rzcF · 2023-11-22
> >
> > Thank you for your detailed answer to my questions.
> >
> > For **Q2** (weak contributions): the authors mention that Theorem 3.5 is their main contribution. I still find it very weak and it follows from existing results in a straightforward way. The proof of Theorem 3.5 is presented in Appendix A and consists in two lemmas:
> >
> > * Lemma A.4 is just the classical lemma that says that Policy Iteration generates a decreasing sequence of value functions, see Proposition 6.4.1 in Puterman.
> >
> > * Lemma A.5 says that the robust Bellman operators are monotonic (non-decreasing) functions. This is also known and widely recognized in the community, see for instance Proposition 6.3.2 in Puterman.
> >
> > * The proof of Theorem 3.5 consists of literally 5 lines of algebra, based on the above two lemmas.
> >
> > Overall I still find the paper contributions to be very weak. The authors should provide appropriate citations for Lemma A.4 and Lemma A.5 (eventually the Puterman textbook), eventually keeping their proof for completeness. The way it is presented right now makes it look like the authors are the first to derive these results, which is clearly not the case.
> >
> > For **Q3** (other uncertainty sets): Thanks for your answer. As far as I understand, your method requires a "closed-form" for the worst-case kernel so that you can exploit this to sample. Apart from KL uncertainty set, the only other closed-form for the worst-case transition kernel that I am aware of is for box uncertainty; see for instance Proposition 3 in [A]. Could your method be applied here?
> >
> >
> > [A] Data Uncertainty in Markov Chains: Application to Cost-
> > Effectiveness Analyses of Medical Innovations, Joel Goh, Mohsen Bayati, Stefanos A. Zenios, Sundeep Singh, David Moore.

---

> > > ### Author Response · Authors · 2023-11-22
> > > **Thank you for your comments**
> > >
> > > Thank you for participating in the discussion and providing additional feedback.
> > >
> > > We respectfully disagree that our lemmas are the same as the classic results in Puterman’s book. First of all, we would like to note that our setting is that of **robust** MDPs. The classical results in non-robust MDPs do not always extend to the robust case (e.g., non-rectangular uncertainty set). In fact, many works in robust MDPs are dedicated to exploring such differences (e.g., the non-convexity of the core optimization problem). Thus, it is necessary to prove it, even though the proof techniques might appear similar. Second, we would like to note that the update process Lemma A.4 is **not** policy iteration. We are updating the transition kernels rather than the policy. While our proof steps bear some resemblance to those of the Prop. 6.4.1 in Puterman, we do not see any direct connections that allow one to trivially deduce Lemma A.4 from Prop. 6.4.1. Finally, we have never claimed that we are the first to derive the classic results in the non-robust MDP setting. We apologize for leaving this impression. We will make the necessary changes to reduce such confusion.
> > >
> > > We note that a similar argument would invalidate much of the results in robust MDPs - for example, the well-cited results of Iyengar or Nilim and El-Ghaoui would also be considered “trivial”.
> > >
> > > We would like to clarify that EWoK relies on the connection between the worst kernel and the nominal kernel, rather than a “closed-form” for the worst kernel. More specifically, EWoK is not directly sampling from the worst kernel but changing the distributions of samples obtained from the nominal kernel. This enables EWoK to scale to high-dimension domains, where we have no access to the worst kernel. The uncertainty set in [A] is structured differently, which does not involve a nominal kernel. As the settings are different, we are afraid that our method could not be directly applied.

---

> ### Comment · Reviewer_rzcF · 2023-11-22
>
> Thanks for your quick response. I do not agree with some of your statements, as I describe below.
>
> * *Difference between nominal MDPs and robust MDPs.* I agree with you that RMDPs and nominal MDPs are different and that in some new settings it is important to reprove properties that may seem obvious for nominal MDPs. However, the setting chosen in this paper is sa-rectangular RMDPs with KL-based uncertainty sets, which has been considered in many other papers. It is thus not necessary to reprove Lemma A.5 (monotonicity of the robust evaluation operator) and the other results from Appendix A.3, it suffices to cite the appropriate references.
>
> * *Lemma A.4 is **not** policy iteration*. I argue that the update (25), which is the focus of Lemma A.4, is *exactly* policy iteration, for the *adversarial MDP*, i.e., for the MDP played by the adversary choosing the transition probabilities $P$. The notion of adversarial MDP is introduced in [1,2,3]. In [3] the authors use policy iteration in the adversarial MDP, see Eq. (7) in [3]. This is exactly the update (25) in your paper.
>
> * *Analogy with the seminal papers.* The results in Iyengar and Nilim El Ghaoui are non-trivial because these authors work with a max-min optimization problem. In the Appendix A.5 of your paper, only the minimization problem is considered, so that this problem falls in the classical MDP theory (though the action set is compact, but this has been studied in prior work as I explain above).
>
> * *Other uncertainty sets.* Thanks for the detailed response.
>
> [1] Vineet Goyal and Julien Grand-Clement. Robust Markov decision processes: Beyond rectangularity. Mathematics of Operations Research, 2022.
>
> [2] Chin Pang Ho, Marek Petrik, and Wolfram Wiesemann. Partial policy iteration for l1-robust Markov decision processes. The Journal of Machine Learning Research, 22(1):12612–12657, 2021
>
> [3] GOH, Joel, BAYATI, Mohsen, ZENIOS, Stefanos A., et al. Data uncertainty in Markov chains: Application to cost-effectiveness analyses of medical innovations. Operations Research, 2018, vol. 66, no 3, p. 697-715.

---

> > ### Author Response · Authors · 2023-11-23
> > **Thank you for your feedback**
> >
> > The reviewer seems to consider our results in Appendix A.3 obvious. While triviality may be in the eyes of the beholder, we believe our work holds empirical merit. From a pedagogical perspective, we still think the results require proof, but if the paper is accepted, we can leave citations instead of the whole proof.

---

### Official Review · Reviewer_QAuy · 2023-10-27

**Soundness:** 3 good
**Presentation:** 2 fair
**Contribution:** 3 good
**Rating:** 6
**Confidence:** 2

**Summary:**

This paper studies online RL in robust Markov decision processes from the perspective of worst transition kernel estimation, which can help scale RMDP-based methods to high-dimensional domains and is thus of great significance to the advance of robust RL. The authors start from the theoretical side by giving a closed form solution to the worst case transition kernel (Theorem 3.2). Motivated by the explicit expression, an approximation of the worst case transition is proposed. The proposed algorithm is extensively studied on various robust RL experimental setups.

**Strengths:**

1. The paper first characterizes the worst case transition dynamic within KL-constrained uncertainty set by a closed form solution (Theorem 3.2), which is of independent interests for future researches on robust MDPs.
2. The idea to simulate the worst case transition based on an approximation of the closed form solution is novel.
3. The method break the curse of scalability of traditional RMDP-based methods with several experimental demonstrations in complex RL domains. The experiments are well organized and convincing.

**Weaknesses:**

1. I think the idea of using a resampling trick to simulate the true worst case transition dynamic (Line 4 of Algorithm 1) is interesting and makes sense given the product form of the solution (Theorem 3.2). However, the intuition behind the empirical choice of the unknown parameter $\omega_{s,a}$ (Eq. 12) is somehow elusive, even the authors provided Proposition 3.4 to argue.
2. Minor typos and notation clarity problem in the theory part (Section 3).

Please see my questions below.

**Questions:**

1. About the Weakness 1 I mentioned, I would appreciate it if the authors could explain more about the empirical choice of the unknown parameter  $\omega_{s,a}$ (since the empirical average over the samples $s_i^{\prime}$ from $\bar{P}(\cdot|s,a)$ forms an overestimation of $\omega_{s,a}$ as suggested by Proposition 3.4).
2. Some typos and notation clarity: (i) in Eq (12) it should be $\sum_{i=1}^Nv(s_i')$; (ii) the notion of $\omega_n$ and $\kappa_n$ is not pre-defined (even with subscript $s,a$). I suggest explicitly giving them a definition (as the parameter associated with the worst case transition kernel when the target function is $v_{P_n}^{\pi}$).

---

> ### Author Response · Authors · 2023-11-15
> **Authors' response**
>
> Thank you for your helpful and valuable comments and feedback. We are thrilled to hear that you find the idea of our method to be novel and that our experiments are well-organized and convincing. Below, we address the comments and questions.
>
> >
> `Question 1 (Q1):` About the Weakness 1 I mentioned, I would appreciate it if the authors could explain more about the empirical choice of the unknown parameter $\omega_{sa}$ (since the empirical average over the samples $s_i'$ from $\bar{P}(\cdot|s,a)$ forms an overestimation of $\omega_{sa}$ as suggested by Proposition 3.4).
>
> `Response 1 (R1):` Thank you for this thoughtful question. The empirical approximation of $\omega_{sa}$ is mainly motivated by implementation considerations that we only have access to samples from the nominal transition kernel. In the future, we will explore more clever ways to approximate it to mitigate overestimation.
>
>
> >
> `Q2:` Some typos and notation clarity: (i) in Eq (12) it should be $\sum^N_{i=1} v(s_i')$; (ii) the notion of $\omega_n$ and $\kappa_n$ is not pre-defined (even with subscript $s,a$). I suggest explicitly giving them a definition (as the parameter associated with the worst-case transition kernel when the target function is $v_{P_n}^\pi$).
>
> `R2:` Thank you for pinpointing those typos and notation issues! We have fixed them in the newest version.

---

> ### Author Response · Authors · 2023-11-22
> **Thank you**
>
> Thank you very much for your valuable feedback. As the discussion period is nearing its end, we kindly ask if you have any additional questions or concerns. Should all the raised issues be satisfactorily addressed, we hope you might consider adjusting the score accordingly. We look forward to your input and insights!

---

> > ### Comment · Reviewer_QAuy · 2023-11-22
> >
> > Thank you very much for your answer to my questions! I have read the responses and I will keep my rating as 6.

---

### Official Review · Reviewer_qiNv · 2023-10-30

**Soundness:** 3 good
**Presentation:** 3 good
**Contribution:** 2 fair
**Rating:** 6
**Confidence:** 4

**Summary:**

This paper proposes a new approach, called EWOK, to address robust MDPs based on the KL-divergence $(s,a)$-rectangular ambiguity set. Specifically, this paper simulates the transited state to approximate the worst-case transition kernel based on the analytical form of the robust Bellman update. Also, the comprehensive experiment illustrates the robustness and good performance of the EWoK in various RL environments.

**Strengths:**

This paper is easy to follow. While EWoK is provided based on specific KL-divergence $(s,a)$-rectangular ambiguity set, it provides a new perspective to estimate the worst-case transitions.

In the experiments, the authors compare their algorithm to both the benchmark non-robust algorithm and the commonly used domain randomization method. The results demonstrate the outperformance of the proposed algorithm in multiple RL practical problems.

**Weaknesses:**

One reason for not giving a higher score at this point is that it seems to me that all the results in this particular paper are rather intuitive or expected. It is worth noting that the convergence analysis of $(s,a)$-rectangular RMDPs has already been extensively studied. While Theorem 3.2 provides the explicit form of the worst-case transition kernel for the KL-divergence ambiguity set, other results do not seem particularly surprising. In particular, Theorem 3.5 is a standard result in the analysis of RMDPs, which seems this paper has limited theoretical contributions.

Another aspect that seems to be lacking is a discussion on how the radius $\beta_{sa}$ of the ambiguity set affects the algorithm's performance, although it would be transferred to new parameters $\omega_{sa}$ and $\kappa_{sa}$. I do expect that the parameter selection procedure could be discussed more.

**Questions:**

The numerical results and theoretical discussion make sense to me. I have the following questions and suggestions:
1. The literature review is not comprehensive. A recent paper [1] also studied RMDPs with global optimality, and it would be helpful if the author discussed it.
2. The official definition of the robust Bellman operator should be added in Section 2.3 for completeness.
3. While we consider a practical problem lying in the KL-based $(s,a)$-
rectangular ambiguity set, the only parameter that the agent can choose is $\beta_{sa}$; however, the other two parameters $\omega_{sa}$ and $\kappa_{sa}$ would be settled directly. Could you explain more about the relationship between $\beta_{sa}$ and the other two parameters $\omega_{sa}$ and $\kappa_{sa}$, or how can the agent reach the latter when setting the former?

[1] Wang, Qiuhao and Ho, Chin Pang and Petrik, Marek. "Policy Gradient in Robust MDPs with Global Convergence Guarantee." ICML (2023)

---

> ### Author Response · Authors · 2023-11-15
> **Authors' response**
>
> We express our gratitude for your valuable suggestions and insightful feedback. It is encouraging to to learn of your appreciation for the clarity of our paper and the effectiveness of our EWoK method. Below, we address the questions and comments.
>
> >
> `Question 1 (Q1):` One reason for not giving a higher score at this point is that it seems to me that all the results in this particular paper are rather intuitive or expected. ... In particular, Theorem 3.5 is a standard result in the analysis of RMDPs, which seems this paper has limited theoretical contributions.
>
> `Response 1 (R1):` To the best of our knowledge, Theorem 3.5 is novel and not straightforward to prove. It establishes the worst kernel can be found iteratively, while each iteration can be estimated via sampling in a model-free setting. We would like to clarify the convergence in Theorem 3.5 is different from the standard convergence analysis of (s,a)-rectangular RMDP. If the reviewer is aware of any existing results akin to Theorem 3.5, we kindly request a reference and will include it to enhance the accuracy of our paper.
>
> >
> `Q2:` Another aspect that seems to be lacking is a discussion on how the radius $\beta_{sa}$ of the ambiguity set affects the algorithm's performance, although it would be transferred to new parameters $\omega_{sa}$ and $\kappa_{sa}$. I do expect that the parameter selection procedure could be discussed more.
>
> `R2:` Thank you for highlighting this point. We would like to clarify that $\beta_{sa}$ is more of a parameter for modeling purposes. In practice, we do not set it directly. In principle, a higher $\beta_{sa}$ means a larger uncertainty set, thus a more conservative policy. However, as all non-tabular experiments never have truly rectangular or KL-constrained uncertainty sets, the influence on practical performance might vary.
>
> >
> `Q3:` The literature review is not comprehensive. A recent paper [1] also studied RMDPs with global optimality, and it would be helpful if the author discussed it.
>
> `R3:` Thank you for pointing out this related work. Similar to our work, it also iteratively computes the worst kernel (appearing in its Algorithm 2). However, their method is gradient-based with iteration complexity of $O(\frac{S^3A}{(1-\gamma)^6\epsilon^2})$ (Theorem 4.4 in [1]). In comparison, our method (update rule in our Equation 14) has the iteration complexity of $O(\log(\frac{1}{\epsilon}))$ as stated in our Theorem 3.5. We have included it in the updated version.
>
> >
> `Q4:` The official definition of the robust Bellman operator should be added in Section 2.3 for completeness.
>
> `R4:` Thank you for your suggestion. We have addressed it in the new version.
>
> >
> `Q5:` While we consider a practical problem lying in the KL-based $(s,a)$-rectangular ambiguity set, the only parameter that the agent can choose is $\beta_{sa}$; however, the other two parameters $\omega_{sa}$ and $\kappa_{sa}$ would be settled directly. Could you explain more about the relationship between $\beta_{sa}$ and the other two parameters $\omega_{sa}$ and $\kappa_{sa}$ , or how can the agent reach the latter when setting the former?
>
> `R5:` Thank you for raising the question. We apologize for the unclarity in our presentation. $\beta_{sa}$ is more of a parameter for modeling purposes, not something chosen by the agent. The connection between $\beta_{sa}$ and the other parameters $\omega_{sa}$ and $\kappa_{sa}$ are described in Eqn.(10) and Proposition 3.3. $\omega_{sa}$ and $\kappa_{sa}$ are the solutions of Eqn.(10), though we can't obtain a closed form for them. In our experiments, we consider $\kappa_{sa}$ as a hyperparameter, and setting it essentially decides a $\beta_{sa}$ for modeling the problem.

---

> ### Author Response · Authors · 2023-11-22
> **Thank you**
>
> Thank you very much for your valuable feedback. As the discussion period is nearing its end, we kindly ask if you have any additional questions or concerns. Should all the raised issues be satisfactorily addressed, we hope you might consider adjusting the score accordingly. We look forward to your input and insights!

---

### Official Review · Reviewer_vpVH · 2023-10-31

**Soundness:** 3 good
**Presentation:** 2 fair
**Contribution:** 3 good
**Rating:** 6
**Confidence:** 3

**Summary:**

The paper aims to solve the Robust Markov Decision Process (RMDP) problem in a realistic high-dimensional scenario, and introduces a method named EWoK. EWoK assigns a higher probability to the transition where the next state has a lower estimated value such that the agent gets a higher chance to learn from the worse transition. EWoK acts as a layer between the true transition metrics and the agent and directly changes the sampled next state, rather than requiring any specific change on the learning agent. Thus, it is able to work with any non-robust reinforcement learning algorithm.

**Strengths:**

- The paper focuses on a nice topic, dealing with the issue that many RMDP algorithms cannot scale to high-dimensional domains. The idea of EWoK is creative. The method bridges robust and non-robust reinforcement learning algorithms, by changing the transition metrics to let it focus more on the robust case and learning the policy with any non-robust algorithm. Therefore, EWoK reserves the ability to scale to high-dimensional inputs by learning a policy using existing reinforcement learning algorithms.

- The experiment section has enough runs (40 and 10 seeds in different environments) to provide relatively reliable average and confidence intervals.

- The experiment section provides ablation studies for two introduced parameters ($\kappa$ and $N$).

**Weaknesses:**

However, the method may need further improvement.

- I am not convinced yet that EWoK can be applied in a realistic domain as claimed. As the paper indicates in the conclusion section, EWoK assumes the environment is able to sample from the same state and action pair multiple times. This requirement is easy to achieve when using simulators or when there exists a perfect model, but is unrealistic in real environments. In the real world, it is almost impossible to reset the environment to the previous state and apply the same action multiple times. Given that the paper defines EWoK as an online method for realistic domains, I think this assumption contradicts the scenario which EWoK is supposed to work with. It might be more accurate if the paper reduces the scope to high-dimensional domains.

- At the end of the experiments section, the paper mentions a larger number of next-state samples does not affect the wall clock time a lot. It is nice to notice and discuss the running time of a method, but  I would like to point out that this happens in simulators because simulators react fast. In real-world scenarios, the environment could be much slower for sampling one next state. It would be nice to also check a case in robotics, or some other environments taking relatively long time to respond.

- It might be worth checking a more difficult experiment setting, such as the one single trajectory case.

Some related works:

Zhou, Zhengqing, et al. "Finite-sample regret bound for distributionally robust offline tabular reinforcement learning." International Conference on Artificial Intelligence and Statistics. PMLR, 2021.

Yang, Wenhao, Liangyu Zhang, and Zhihua Zhang. "Toward theoretical understandings of robust Markov decision processes: Sample complexity and asymptotics." The Annals of Statistics 50.6 (2022): 3223-3248.

Yang, Wenhao, et al. "Robust Markov Decision Processes without Model Estimation." arXiv preprint arXiv:2302.01248 (2023).

**Questions:**

- Could the authors provide a learning curve to show how long the method takes to converge? It would be better if the learning curve plot could also include your baselines.

---

> ### Author Response · Authors · 2023-11-15
> **Authors' response**
>
> We extend our gratitude for your insightful and constructive feedback. We are happy to see your comment that the idea of EWoK is creative. Below, we address the questions.
>
> >
> `Question 1 (Q1):` I am not convinced yet that EWoK can be applied in a realistic domain as claimed. ... It might be more accurate if the paper reduces the scope to high-dimensional domains.
>
> `Response 1 (R1):` Thank you for the suggestion. While EWoK shows potential for integration with learned models to get rid of the assumption, we concur that it is more accurate to reduce the current scope to high-dimensional domains. We have revised the paper to embody this change.
>
> >
> `Q2:` At the end of the experiments section, the paper mentions a larger number of next-state samples does not affect the wall clock time a lot. ... It would be nice to also check a case in robotics, or some other environments taking relatively long time to respond.
>
> `R2:` Thank you for highlighting this point. We agree that the negligible overhead is due to fast simulation, which may not be available in real-world scenarios. We have revised our paper to reduce the confusion. In real-world cases where sampling next states is time-consuming, it could increase simulation time but we believe that this increase won't be prohibitively expensive in most realistic cases.
>
> >
> `Q3:` It might be worth checking a more difficult experiment setting, such as the one single trajectory case.
>
> `R3:` Thank you for the suggestion. Could you provide more details about this specific setting? We would gladly conduct the requested experiments.
>
> >
> `Q4:` Could the authors provide a learning curve to show how long the method takes to converge? It would be better if the learning curve plot could also include your baselines.
>
> `R4:` Sure, we have updated the paper and included the learning curves in the appendix (Fig. 13). Thank you for pointing it out.

---

> ### Author Response · Authors · 2023-11-22
> **Thank you**
>
> Thank you very much for your valuable feedback. As the discussion period is nearing its end, we kindly ask if you have any additional questions or concerns. Should all the raised issues be satisfactorily addressed, we hope you might consider adjusting the score accordingly. We look forward to your input and insights!

---

> > ### Comment · Reviewer_vpVH · 2023-11-22
> > **Reply**
> >
> > I would like to thank the author for the reply. I will keep my score.
> >
> > For Q3, I was thinking of an environment that can only provide a long and consecutive trajectory. This is to test how the result looks like in a more realistic case, or in an extreme case for the proposed algorithm, i.e. there is no resetting on state s_t and N=1. This experiment can act as part of the parameter study, also tell us how the algorithm might perform in reality. But as the scope has already been modified, I do not consider this test for realistic cases as important as before.

---

> > > ### Author Response · Authors · 2023-11-23
> > > **Thank you for your reply**
> > >
> > > Thank you for your clarification. This is an interesting case. When $N=1$ (without resetting), the sample distribution will follow the nominal kernel. One idea we can think of is to learn a transition model and obtain additional fictitious samples from it. We will delve deeper into this direction in our future work.

---

### Meta-Review · Area_Chair_2bc5 · 2023-12-11

**Metareview:**

This paper presents an interesting idea to turn any non-robust RL algorithm into a robust RL method. This is accomplished by estimating the worst-case transition dynamics from the environment and simulating experience from that model. This should allow for the extension of robust MDP methods to higher dimensional settings, and there are experiments on standard RL benchmarks (with modified parameters) to show that the agent’s performance is slightly more robust than SAC or domain randomization. While the idea is promising, the technical contributions (theoretical and empirical) are not as impressive. Additionally, the experiments only show a minor impact on performance in many settings, making it unclear how much the policy changes in response to the worst-case kernel. The choice of the interquartile mean as a performance metric is also peculiar as it throws out the worst 25% of runs. As such, I do not recommend this paper for acceptance.

**Justification For Why Not Higher Score:**

While the idea in this paper is interesting, the technical contributions are lacking. Either strong theoretical results are needed or more detailed experiments to show an understanding that the method is actually providing the stated robustness.

**Justification For Why Not Lower Score:**

N/A

---

### Decision · Program_Chairs · 2024-01-16

Reject